# A three-membered ring approach to carbonyl olefination

Supaporn Niyomchon[1], Alberto Oppedisano[1], Paul Aillard[1] & Nuno Maulide [1]

The carbon–carbon double bond, with its diverse and multifaceted reactivity, occupies a prominent position in organic synthesis. Although a variety of simple alkenes are readily available, the mild and chemoselective introduction of a unit of unsaturation into a functionalized organic molecule remains an ongoing area of research, and the olefination of carbonyl compounds is a cornerstone of such approaches. Here we show the direct olefination of hydrazones via the intermediacy of three-membered ring species generated by addition of sulfoxonium ylides, departing from the general dogma of alkenes synthesis from carbonyls. Moreover, the mild reaction conditions and operational simplicity of the transformation render the methodology appealing from a practical point of view.

[1] Faculty of Chemistry, Institute of Organic Chemistry, University of Vienna, Währinger Strasse 38, 1090 Vienna, Austria. Correspondence and requests for materials should be addressed to N.M. (email: nuno.maulide@univie.ac.at)

The C–C double bond is a functional group of central importance in organic chemistry, it is expressed as such in countless secondary metabolites and, more importantly, it is arguably one of the most useful functionalities for the practitioners of the art of total synthesis[1]. Some of the most powerful reactions available to organic chemists rely on the reactivity of olefins: the Heck reaction, olefin metathesis and the Sharpless asymmetric dihydroxylation are notable examples[2–4], coincidentally all recognized by Nobel Prizes during the last 16 years. There are a number of different possible approaches for the introduction of a double bond into a molecule, amongst which the olefination of carbonyl moieties is the most developed[5, 6]. Since the 1950s and the landmark report of the Wittig reaction[7], a multitude of variations and new methodologies have been developed, such as the Peterson, Julia, and Tebbe olefinations or the HWE modification of Wittig's original conditions[7–11]. Notwithstanding, investigation into the improvement and expansion of the toolset of reactions employed to introduce olefins into molecules remains a rich field of research to this day[12–14]. With this idea in mind, we embarked on the development of a strategy to achieve the olefination of carbonyls.

Typical classifications of olefination methodologies tend to discriminate between the nature of the reagent employed, such as phosphonium ylides (Wittig), sulfones (Julia), silicon-stabilized carbanions (Peterson), or metal alkylidenes (Tebbe). Given that all these reactions are assumed to proceed through cyclic intermediates which undergo different types of cycloreversion reactions in the olefin-yielding step, we propose a different and perhaps richer view, assigning a classification based on the ring size of that key reaction intermediate (Fig. 1a). For instance, the modified Julia (so-called Julia–Kocienski) reaction involves a five-membered spirocyclic intermediate, whereas the Wittig, Tebbe, and Peterson olefinations all proceed via a four-membered ring, as does recently developed azaphosphetanes chemistry[15, 16]. The conspicuous scarcity of three-membered ring intermediates in this analysis drove us to develop a new approach to olefination relying on an aziridine intermediate[17–20].

**Table 1 Optimization of reaction conditions**

1a R¹ = CO₂Me, R² = Me 2a R¹ = CO₂Me, R² = Me 90%
1b R¹ = R² = Ph 2b R¹ = R² = Ph 92%
1c R¹ = R² = Me 2c R¹ = R² = Me 90%

| Entry | Azine | Pronucleophile | Base | Temperature (°C) | Isolated yield[a] |
|-------|-------|----------------|------|------------------|-------------------|
| 1 | 2a | BrCH₂COPh | LiHMDS | −78 | 0% |
| 2 | 2a | CH₂Br₂ | n-BuLi | −78 | 0% |
| 3 | 2a | Me₃S(I) | NaH | −20 | 9% |
| 4 | 2a | Me₃SO(I) | NaH | 0 | 71% |
| 5 | 2a | Me₃SO(I) | t-BuOK | r.t. | 84% |
| 6 | 2a | Me₃SO(I) | t-BuONa | r.t. | 60% |
| 7 | 2a | Me₃SO(I) | t-BuOLi | r.t. | 27% |
| 8 | 2b | Me₃SO(I) | t-BuOK | r.t. | 53% |
| 9 | 2c | Me₃SO(I) | t-BuOK | r.t. | <10% |
| 10[b] | 2a | Me₃SO(I) | t-BuOK | r.t. | 16% |
| 11[c] | 2a | Me₃SO(I) | t-BuOK | r.t. | 56% |

r.t. room temperature
[a]Isolated yield over two synthetic steps
[b]1.05 equiv. Me₃SO(I), 1.05 equiv. t-BuOK
[c]No exclusion of water

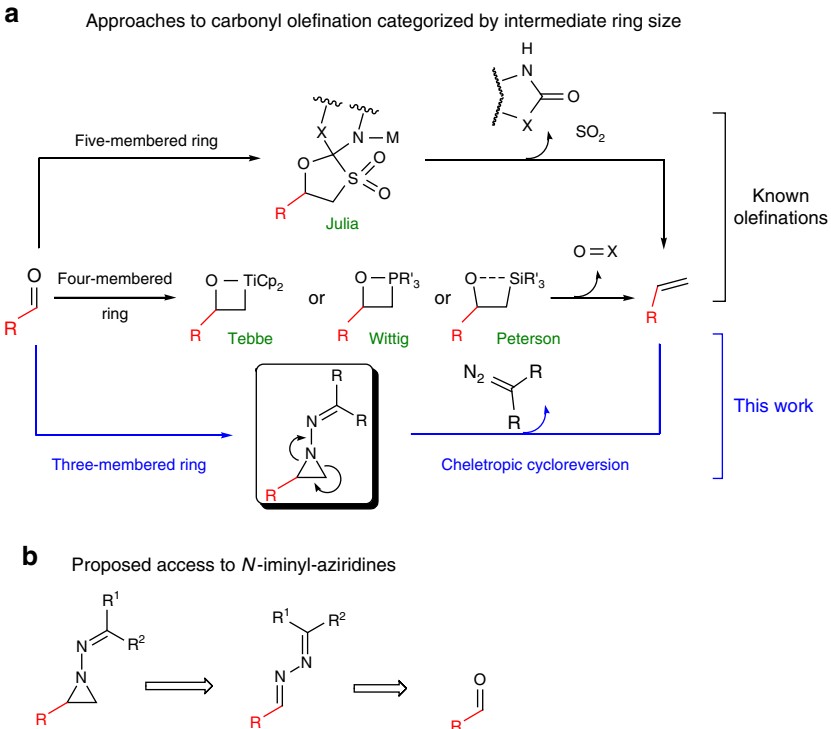

**Fig. 1** Approaches to carbonyl olefination. **a** Carbonyl olefination strategies and **b** proposed approach via a three-membered ring intermediate

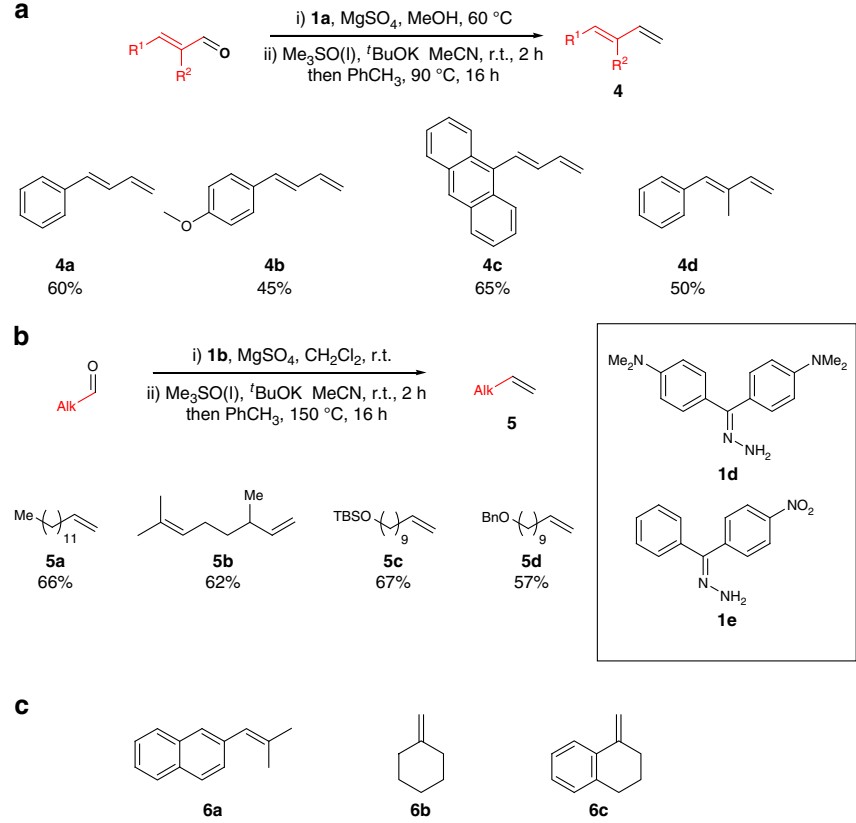

**Fig. 2** Substrate scope of olefination for various aryl-aldehydes. Yields were calculated on isolated yield over two steps. r.t. = room temperature. [a]Low yield due to difficulties in isolation

**Fig. 3** Substrate scope of olefination for various aldehydes. **a** Substrate scope of olefination for enals. Yields calculated on isolated material over two synthetic steps, r.t. = room temperature. **b** Substrate scope of olefination on alkyl-substituted aldehydes. **c** Unsuccessful synthesis of internal olefins

Over the past few years, our group has reported unprecedented transformations that take advantage of underdeveloped reactivity[21, 22]. In this context, we herein present an unusual and conceptually method for the synthesis of olefins from aldehydes, which we believe brings further diversity to the field while possessing synthetic advantages in its own right.

## Results

**Initial considerations.** Our approach, depicted in Fig. 1b predicated access to an olefin from an aldehyde in a unique process. We proposed the transient generation of an *N*-iminyl aziridine and its subsequent cheletropic cycloreversion[23–28] to unveil the desired olefin product.

At the outset of the project, we foresaw two main challenges. First, the formation of an *N*-iminyl aziridine from a carbonyl precursor in a synthetically useful manner[29, 30] and then finding appropriate conditions for a facile cheletropic elimination[31].

We eventually settled for the aziridination of an in-situ generated *N*-iminyl imine (azine) as route to the three-membered ring intermediate (Fig. 1b). Key to addressing the two aforementioned challenges would be the nature of the hydrazone $R^1$ and $R^2$ substituents.

**First experiments**. In initial experiments, we sought to convert 2-napthaldehyde as model substrate to the corresponding azine. We started our investigation with the use of the hydrazone derived from methyl pyruvate (**1a**, Table 1). A facile condensation between the two delivered the crude azine (**2a**) in analytically pure form, on which we explored suitable conditions for aziridine formation. The use of an α-bromo enolate in imino-Darzens-like conditions, as well as the combination of $CH_2Br_2$ and butyl lithium were attempted, but without success (Table 1, entries 1 and 2). We next turned our attention to the use of sulfur ylides, generated in situ from sulfonium salts, as nucleophiles[32]. To our delight, the desired terminal olefin was directly isolated in 9% yield (Table 1, entry 3). Encouraged by this result we turned to sulfoxonium ylides, known to exhibit greater stability, which allowed us to increase the temperature of the reaction[33]. The use of NaH as base led to the formation of the olefin in 71% yield (Table 1, entry 4). Variation of the base and the counterion helped us to identify *t*-BuOK as the best combination (Table 1,

entries 5–7). Changing the substituents $R^1$ and $R^2$ on the imine moiety did not lead to further improvement (Table 1, entries 8 and 9). Finally, a slight excess of sulfoxonium ylide is necessary to achieve high yield (Table 1, entry 10) and rigorous exclusion of water while beneficial is not mandatory (Table 1, entry 11).

**Substrate scope**. With suitable conditions for this olefination in hand, we next explored the scope of aromatic aldehydes. As shown in Fig. 2, a diverse array of electron-neutral (**3a**, **3b**), electron-poor (**3c**–**3f**), and electron-rich (**3g**–**3i**), in addition to heteroaromatic aldehydes (**3j**–**3m**), performed well in this protocol. Due to the mildness of the reaction conditions, a wide range of functional groups are well tolerated: esters, nitro groups, amides, ethers, and aryl halides. Importantly, although the overall transformation implies one extra step for *N*-iminyl imine preparation, this is a very facile operation for all aldehydes studied. Indeed, simple stirring at room temperature in presence of $MgSO_4$ leads to quantitative conversion into the free-flowing, generally yellow powder products. The only side products, detected in the crude products mixtures, are the homologated azines.

We next applied our reaction conditions to α,β-unsaturated azines, derived from enals (Fig. 3a). Surprisingly, the reaction of sulfoxonium ylides with unsaturated azines resulted in a clean 1,2-addition to generate dienes **4** after cheletropic elimination. Notably, no product of conjugate addition was observed by $^1$H NMR analysis, despite the fact that reaction between sulfoxonium ylides and α,β-unsaturated carbonyls is a textbook transformation. This effectively allows a smooth access to synthetically useful *E*-configured di- and trisubstituted dienes (**4a**–**4d**).

The generality of the reaction was also investigated on aliphatic aldehydes (Fig. 3b). In this case, the use of hydrazone **1b** was crucial, as when **1a** was employed, a rapid isomerization to an unstable side product (tentatively assigned as the *N*-iminyl enamine tautomer) was observed. It is likely that **1b**, carrying an extended π-system, stabilizes the hydrazone tautomer. Other types of highly conjugated hydrazones with more electron rich system (**1d**) or more electron poor system (**1e**), were tested but failed in providing the desired product. Compound **1e** exhibited prohibitively slow rate of azine formation and provided trace amounts of olefin with considerable degradation. Compound **1d** led to efficient azine formation, but nucleophilic attack from the sulfoxonium ylide was not observed. Finally with the use of hydrazone **1b**, the desired olefins could be isolated in good yields after cheletropic extrusion triggered by heating in toluene (**5a**, **5b**). Moreover, the reaction conditions tolerate standard protecting groups such as silyl ether- (**5c**) and benzyl (**5d**). To test the limit of applicability of this olefination procedure (Fig. 3c), we

**Fig. 4** Synthesis of internal olefins. Yields are calculated on isolated material over two synthetic steps. See the Methods section for conditions A and B

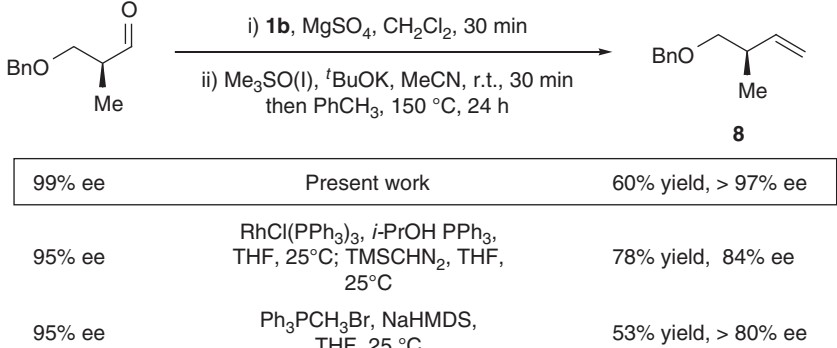

**Fig. 5** Study of the racemization at α-position of the aldehyde. Yields are calculated on isolated material over two synthetic steps

| | | |
|---|---|---|
| 99% ee | Present work | 60% yield, > 97% ee |
| 95% ee | RhCl(PPh₃)₃, *i*-PrOH PPh₃, THF, 25°C; TMSCHN₂, THF, 25°C | 78% yield, 84% ee |
| 95% ee | Ph₃PCH₃Br, NaHMDS, THF, 25 °C | 53% yield, > 80% ee |

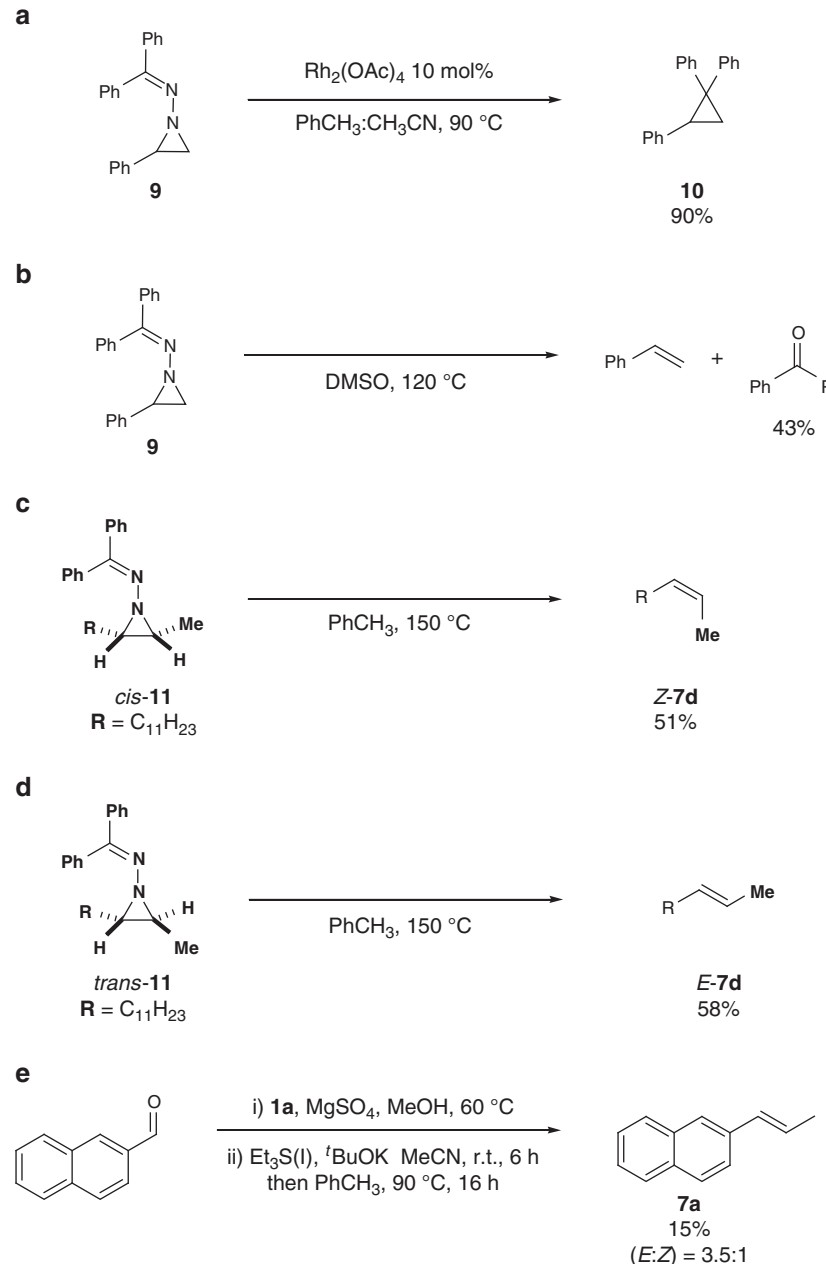

**Fig. 6** Experimental evidences to determine the mechanism of the olefination reaction. **a** Rhodium trapping of the diazo intermediate. **b** Azine decomposition study. **c** Stereochemical outcome from *cis*-azine. **d** Stereochemical outcome from *trans*-azine. **e** Olefination reaction with sulfonium ylide

applied the reaction conditions to form a trisubstituted olefin starting from either a branched sulfoxonium salt (**6a**) or from a ketone (**6b**, **6c**), but no aziridine formation could be observed in any of these examples.

Finally, we investigated the applicability of the methodology to the formation of disubstituted olefins (Fig. 4). By employing appropriately substituted and readily accessible sulfoxonium salts, we were able to obtain internal olefins in good to high yields, starting from aromatic aldehydes (**7a**, **7b**, **7c**) or aliphatic and conjugated aldehydes (**7d**, **7e**). Notably, the alkene products were obtained with marked *E* selectivity. This might be a consequence of either stereospecific cheletropic elimination from the more stable *trans*-aziridine intermediate or non-concerted ring-opening pathways that allow isomerization to take place[23–34]. This *E*-selectivity is a noteworthy trait of the method.

Significantly, this olefination can also be applied to sensitive aldehyde substrates bearing a stereogenic center at the α-position (Fig. 5). It should be noted that, in contrast to Wittig procedures, which usually lead to a significant erosion of enantiopurity by base-mediated epimerization[35], the procedure reported herein produced the desired olefin **7** with minimal loss of chiral information.

**Mechanistic experiments**. Finally, we performed control experiments to propose a reasonable mechanism. *N*-iminyl aziridine **9** (generally prepared and used in situ throughout this manuscript) was isolated and subjected to specific conditions: upon heating **9** to 90 °C in toluene, in presence of catalytic Rh₂(OAc)₄, cyclopropane **10** was isolated in 90% yield (Fig. 6a). When the reaction is performed in DMSO, benzophenone is

**Fig. 7** Proposed mechanism. Proposed transition states for the nucleophilic addition event (**I** and **III**) and depiction of the stereospecific cheletropic elimination (**II** and **IV**) or ionic elimination (**V**).

obtained in 43% yield (Fig. 6b)[36]. These results provide evidence for the formation of a diazo compound through cheletropic elimination. Further proof of a concerted cheletropic elimination can be inferred from the high sterospecificity of the reaction. Pure samples of either *cis* or *trans*-aziridine were subjected to the reaction conditions, delivering respectively pure *Z* (Fig. 6c) or *E* (Fig. 6d) olefin as product. Finally, a different nucleophile for the aziridine formation was used, namely triethyl sulfonium iodide, and a different ratio in the double-bond geometry was obtained (Fig. 6e). Since the aziridine eventually formed would be the same for the two different nucleophiles, and the cheletropic reversion from the latter is stereospecific, a different ratio in the olefin product geometry hints at the fact that the nucleophilic attack on the azine is the stereo-determining step.

Based on these experiments, we are able to propose a working mechanism for the reaction (Fig. 7). Starting from the azine (**2d**), an initial nucleophilic attack of the sulfoxonium ylide on the aldimine carbon delivers a zwitterionic compound, which then quickly collapses in an intramolecular fashion to form the aziridine with extrusion of a sulfoxide molecule. This is the key stereo-determining moment: two possible faces of attack are feasible, aziridine formation via **I** would deliver a *cis* three-membered ring **II** while an attack via **III** would deliver *trans* three-membered ring **IV**. Moreover, the formed aziridines are stable and not prone to geometric interconversion[37, 38]. At this stage is not possible to state the factors leading to a preference for attack on one of the two faces, most probably subtle stereoelectronic factors are playing a role. Eventually, the aziridine, when heated cycloreverts to unveil the desired olefin in a concerted mechanism and in a stereo-specific manner. A two-step ionic mechanism of degradation via **V** is operating at the same time leading to a homologated compound **VI**. Anyway, this is not a reversible process as no geometry swapping is observed.

In conclusion, we have developed a new concept for carbonyl olefination, relying on formation of a three-membered ring as key intermediate. This intermediate, an *N*-iminyl aziridine, is conveniently accessed in a one-pot procedure by addition of a sulfoxonium ylide to an azine, the thermal decomposition of which leads to formation of the desired olefin in good to high yields. Notably, this methodology selectively delivers *trans*-disubstituted olefins and affords dienes from α,β-unsaturated carbonyls, in contrast to the usual selectivity of sulfoxonium ylides. Importantly, α-chiral aldehydes can be olefinated with minimal epimerization. While we are well aware of the power and historical weight of the venerable Wittig, Julia, Peterson, Tebbe, and related olefination procedures, developed continuously over

the past 60 years, we believe that the approach reported herein has significant complementarity to these methods.

## Methods

**Representative procedure for the olefination reaction.** Method A: A mixture of aldehyde (1.0 equiv.) and hydrazone **1a** (1.1 equiv.) was stirred in the presence of MgSO$_4$ (100 mg/mmol) in MeOH at 60 °C overnight. Then, the reaction mixture was filtered and the solvent was removed under reduced pressure. In another flask, potassium *tert*-butoxide (2.0 equiv.) was added to a solution of trimethyl sulfoxonium iodide (2.5 equiv.) in MeCN (0.2 M). The resulting mixture was stirred for 30 min at room temperature. To this solution was added the previously formed azine in MeCN (0.2 M) and the reaction mixture was stirred at room temperature until complete conversion (usually 3–6 h). Toluene (0.2 M) was then added and the reaction mixture was heated to 90 °C overnight. The reaction was then diluted with a saturated aqueous solution of NH$_4$Cl and extracted with EtOAc. The combined organic layers were washed with brine, dried over anhydrous Na$_2$SO$_4$ and concentrated under reduced pressure. The crude product was purified by flash column chromatography on silica gel to afford the desired product.

Method B: A mixture of aldehyde (1.0 equiv.) and hydrazone **1b** (1.1 equiv.) was stirred in the presence of MgSO$_4$ (100 mg/mmol) in CH$_2$Cl$_2$ (0.3 M) at room temperature for 30 min. Then, the reaction mixture was filtered and the solvent was removed under reduced pressure. In a pressure vial, potassium *tert*-butoxide (1.5 equiv.) was added to a solution of trimethyl sulfoxonium iodide (1.75 equiv.) in MeCN (0.3 M). The resulting mixture was stirred for 30 min at room temperature. To this solution was added the previously formed azine in MeCN (1 M) and the reaction mixture was stirred at room temperature until complete conversion. At completion of the reaction, the volatiles were removed under reduced pressure and the solid residues were diluted with toluene (0.5 M) and heated to 150 °C. After 24 h, the reaction is generally complete and the mixture is cooled to room temperature. The solution is then diluted with aqueous NH$_4$Cl and extracted with EtOAc (3×). The combined organic layers were washed with brine and dried over MgSO$_4$. The crude product is then purified via flash chromatography on silica gel to afford the desired product.

**Data availability**. The authors declare that the data supporting the findings of this study are available within the paper and its Supplementary Information Files.

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

## Acknowledgements

We are grateful to the University of Vienna for continued support of our research. Financial support by the European Research Council (StG 278872 FLATOUT), the DFG, and Covestro AG is acknowledged

## Author contributions

N.M. and S.N. conceived and designed the research. S.N., A.O., and P.A. carried out experiments. N.M., S.N., A.O., and P.A. wrote the paper.

## Additional information

**Competing interests:** The authors declare no competing financial interests.

