## [Peer Review File · Nature Communications]

Reviewers' comments:

Reviewer #1 (Remarks to the Author):

The authors have developed a novel method for olefin formation starting from various aldehydes. The aldehyde reacts with a hydrazone to form an azine, which is then reacted with a sulfoxonium ylide to yield an N-iminyl aziridine. The aziridine undergoes cheletropic elimination of a diazoalkane to give alkene product. Mono and di-substituted alkenes were synthesized in moderate to high yields. The method is tolerant of amides, ethers, esters, alkenes (despite propensity of enals themselves to undergo conjugate addition with sulfoxonium ylides), nitro group, diphenylamino, and aryl halides.

Various aldehydes were employed – aromatic aldehydes with different electron withdrawing & donating substituents, enals, and some representative aliphatic aldehydes, including one with a stereogenic centre α to the carbonyl. Essentially no loss of enantiopurity at the stereogenic centre was observed in the latter reaction. Since C–C and C=C bond forming reactions are of paramount importance in organic chemistry, this submission seems to me to be a valuable addition to the existing bond forming methodologies, and improves on the state of the art by addressing certain shortcomings of other methods that nominally achieve the same C=C bond formations.

The authors claim that their method is the first example of a novel concept – olefination of a carbonyl compound via a 3-membered ring intermediate (as opposed to well-known methods that involve 4- and 5-membered intermediates). However, the numerous variations on the Barton-Kellogg reaction, known for approximately 50 years, are based on exactly this concept, with the 3-membered ring in this case being a thiirane. The thiirane can be formed starting from an azine + hydrogen sulphide, or from a diazoalkane + thiocarbonyl compound. The azine and the diazoalkane, in the different variants, are formed from carbonyl compounds. Extrusion of sulphur from the thiirane is brought about by various means. Treatment with a phosphine also yields the olefin product, presumably through a thiaphosphetane intermediate (i.e. in this instance, it is a 4-center process). There are obvious parallels between the above precedents and the method in the submitted article – certainly, reference should be made to these precedents, with the addition of several literature citations. Most of the salient references can be found in OBC 2005, 3, 28. Other relevant references include Chem. Commun. 1972, 4, and references therein. Since there is clear precedent for the concept of olefination of a carbonyl group through a 3-membered ring intermediate, the authors should acknowledge this.

It would also be appropriate to add references to several papers of Shi-Kai Tian to the introduction, e.g. Chem. Commun. 2011, 47, 2158; JACS 2010, 132, 5018; EJOE 2011, 1084.

Although there are several similarities to the Barton-Kellogg method, the present method is nonetheless an ingenious departure from the existing state of the art in olefin formation in terms of scope and the exact means through which olefins are formed. Its reasonably general applicability adds significantly to this value. The present method is complementary to the Barton-Kellogg method for olefin formation in the sense that it can be readily used to produce mono or disubstituted olefins, which may not be possible with the Barton-Kellogg method. It may also be complementary to the Wittig reaction (although more examples would be needed to show this unequivocally) in that it appears to yield E-alkenes with alkyl substituents where the Wittig reaction (as it is typically applied) would give Z-alkenes.

In order to increase the general applicability of the method, it would be interesting to see at least one example of formation of a trisubstituted alkene, perhaps by using a sulfoxonium ylide with a branched nucleophilic carbon. I suggest attempting this reaction and including the results of the experiments (whether successful or not) in the present paper. An attempt should also be made to synthesize a disubstituted alkene starting from an aliphatic aldehyde (i.e., Fig 5 with an aliphatic aldehyde).

The authors refer to optimisation of the reaction conditions (Table 1 of article). It is not entirely clear to me why more than 1 equivalent of the trimethylsulfoxonium ylide should be required (i.e. it seems in principle that 1.05 equivalents of sulfoxonium salt + 1.0 equivalent of base should suffice). If the authors have not tried this set of conditions, they should do so and include details of their observations in the article. If reactions under these conditions have already been attempted, details of the results should be included – in particular if there is a decrease in the yield.

For disubstituted olefins (Fig 5), was the ratio of the isomers checked prior to subjecting the product to column chromatography? Isomerization seems unlikely for the alkenes synthesized in this article, but it would be worth checking the ratio of the isomers in the crude product prior to chromatography to be sure that the ratios quoted really are indicative of the selectivity from the reaction rather than an artefact of how the product was purified.

Was there any evidence of addition of the sulfoxonium ylide to the more substituted end of the azine? Particularly in cases where the yields were low? In cases in which low yields were obtained, what was the major side-product? Was there a lot of starting material remaining? What caused the low yields in those cases – e.g. any of the entries of Fig 3? Please insert comments on the identities of side products, or if apparently unreacted starting material remained at the end of the reactions that gave relatively low yields.

The authors should include a diagram and some discussion in which it is explicitly shown that the initial cheletropic elimination product is diazoalkane, and in which an explanation is given for what happens to this product subsequently. This information, implied in the submitted article, may not be obvious to readers who are not familiar with the area.

Comments on Mechanistic Study

Unless I am mistaken, no details of the experiment in which 8 is reacted with DMSO are included in the experimental section. A description of this experiment with NMR spectra should be included. What products, aside from the 43% that formed benzophenone, were formed in this process?

The mechanism of the decomposition of 8 needs further clarification – the evidence already presented, although consistent with the authors' rationale, is not sufficiently comprehensive. The yield of the reaction of 8 with DMSO is somewhat low and, moreover, no details are given about this experiment in the supporting information. Use of commercially available 4,4'-bis(N,N-dimethylamino)benzophenone in a similar reaction sequence to that given in the submission would lead to an iminyl aziridine and hence a diazo diarylmethane with two very electron rich aryl groups, which may give a high yield in an oxidation reaction with DMSO based on reference 31 from the submitted article. Alternatively, 4,4'-bis(methoxy)benzophenone could be used. A high yield in a reaction such as this would show that a diazoalkane is indeed formed by cheletropic elimination from the iminyl aziridine.

Is there evidence for the formation of dimethylsulfide as by-product in the reaction of 8 with DMSO? If not, what else is formed as a by-product?

Comments on Reactions with Aliphatic Aldehydes

It is stated on page 7 of the article that "It is likely that 1b, carrying an extended π -system, stabilizes the hydrazone tautomer". Use of hydrazones derived from substituted benzophenones to form azines derived from aliphatic aldehydes might allow the above supposition to be tested.

An azine derived from commercially available 4-nitrobenzophenone and an aliphatic aldehyde should favour the hydrazone tautomer, and thus lead to higher yields in reactions with aliphatic aldehydes. The authors should consider investigating this possibility.

An azine derived from 4,4'-bis(N,N-dimethylamino)benzophenone and an aliphatic aldehyde should favour the enamine tautomer due to the strongly electron donating character of the dimethylamino groups.

Comments on Supporting Information

The numbering of the iminyl aziridine, and the cyclopropane product (8 and 9, respectively, in the main article) is not consistent between the main article and supporting information – see pg 14 of Supporting Information.

Synthesis of hydrazone 1b (pg 4) – why does the description say “ketone or aldehyde”? Surely only benzophenone was used?

1b is incorrectly labelled as a hydrazine rather than a hydrazone in the supporting information (pg 4)

The reactions were all carried out under anhydrous conditions. How exactly was the filtration of the azine solution carried out? If it was done under inert atmosphere, please provide more detail on the method of filtration used.

To what extent do the authors think water affects the yields of the reactions? It is clearly a factor, since the reactions were carried out under anhydrous conditions. Could the lower yields in some cases be a result of the intervention of water? Perhaps activated molecular sieves could be used in place of MgSO₄ to increase yields – 3 Å sieves are suitable for drying acetonitrile (see JOC 2010, 75, 8351).

Does the formation of the azine have to be done in CH₂Cl₂? Perhaps it could be done in CH₃CN, followed by filtration to remove MgSO₄, removing the need for solvent removal after formation of the azine. If this is not possible, the authors should comment on why the use of CH₂Cl₂ for the azine formation is essential, either in the Supporting Information or in the main article.

Comments on the Stereoselectivity of Disubstituted Alkene Formations

Significant elaboration on the reasons for the observed E-stereoselectivity are required. At the outset, and as stated elsewhere, the isomer ratios of the crude products should be checked. The authors state that the E-selectivity may be a result of “stereospecific cheletropic elimination of the more stable trans-aziridine”. At the very least, this sentence needs to be re-formulated, as it implies that the aziridine itself is eliminated (i.e., the phrasing should be made more precise). However, a more serious issue is what the sentence implies about the chemistry. Do the authors think that the formation of the iminyl aziridine is reversible, and that E-selectivity results from elimination from the trans-aziridine being faster than elimination from the cis-aziridine? Or is the formation of the aziridine irreversible, meaning that the predominant E-selectivity is due to faster formation of the E-aziridine? It may be difficult to answer these questions (although the experiments suggested below would help), but a more precise explanation of what the authors believe to be possible is warranted.

Given that it has been shown to be possible to isolate 8, perhaps the authors should attempt to isolate an iminyl aziridine that goes on to give a vicinally disubstituted alkene (e.g. derived from a triethylsulfoxonium salt and t-BuOK), check its stereochemistry, and then check if the stereochemistry is retained (or otherwise) during the course of decomposition to alkene. This would give a wealth of information about the concertedness, or otherwise, of the second step of the reaction, and about the source of the stereoselectivity. Trapping experiments could also be employed to check if the elimination is not concerted.

There are several examples of typographical errors, for example

Phosphorus – main article pg 13

triethylsulfoxonium iodide - ESI pg 13

Please fix all such errors. Use “room temperature” in place of rt in main article and supporting information. Also “crude” is not a noun – use “crude product” (experimental sections).

Recommendation

I recommend that this article be accepted once the revisions and additional experiments suggested above have been done. Even if the suggested experiments do not yield the desired results, they may lead to a significantly better understanding of the reaction, its mechanism, and the source of the E-selectivity (where appropriate).

Reviewer #2 (Remarks to the Author):

This manuscript by Maulide and co-workers describes a method for olefination of aldehydes via sequential N-iminyl aziridine formation followed by cheletropic cycloreversion. The proposed mechanism involves formation of a three-membered aziridine intermediate using sulfoxonium iodide as the methylidene source with subsequent thermal-induced elimination of a diazo species to reveal the olefin.

Conceptually, the reaction is interesting, as it represents a new mechanistic strategy for carbonyl olefination that is distinct from the traditional approaches in Figure 1. Practically speaking, the reaction uses straightforward reaction conditions, the yields are typically good, and the functional group tolerance is reasonably well established. There are some notable limitations of the method. First, the substrate scope appears to be somewhat narrow; yields with aliphatic aldehydes are in the 50–70% range, and no examples are provided with ketones to prepare 1,1-disubstituted olefins. Only two examples are provided of non-methylidene alkylidene transfer to prepare 1,2-disubstituted alkenes. In the context of multi-step organic synthesis, it is unclear what advantages this method might offer compared to existing methods.

The Supporting Information is well organized, detailed, and complete. All new compounds are thoroughly characterized. Aside from the points below, the conclusions are generally supported by the data, and the manuscript is presented in a scholarly manner. Based on the overall conceptual novelty, I recommend that this manuscript be published in Nature Communications, following incorporation of the following minor revisions:

(1) For examples of non-methylidene alkylidene transfer to prepare 1,2-disubstituted alkenes, the levels of E selectivity appear to be variable based on the two examples that are provided (>20:1 in one case, 4:1 in the other). More 1,2-disubstituted examples are necessary to make general conclusions about the stereoselectivity of this transformation.

(2) In Ref. 30, which is cited in the text accompanying Fig. 6, the two Wittig-type examples that the authors refer to with this specific substrate involve epimerization at the level of 95% to 84% ee and 95% to 80% ee. The authors describe this as “significant erosion”. Given that the descriptor ‘significant’ is subjective in this context—I imagine some chemists would also call this “modest epimerization”—I would prefer that the actual numbers from this earlier report be presented here to make comparison more straightforward for the reader.

(3) In the figures, it is unclear at first-glance whether the yields correspond to final yield over all three steps or only over the final two steps. From the way that the experimental procedure is written, the former appears to be the case, but it would be helpful to the reader if this were clarified in the footnotes.

(4) A few minor typographical, stylistic, and grammatical issues were present throughout the manuscript and Supporting Information

- Line 8: “an unsaturation” is ambiguous in this context. There may be a missing word here.
- Line 47: insert a comma after ‘project’

- Line 53: '2-napthaldheyde' should be '2-napthaldehyde'
 - Line 69: there should be a hyphen between 'electron' and 'neutral'
 - Line 71: there should be a comma after 'conditions'
 - Line 78 and 79: 'unsatured' should be 'unsaturated'
 - Line 81: suggested revised wording, here: "...analysis, despite the fact that reaction between sulfoxonium ylides and α,β -unsaturated carbonyls is a textbook transformation."
 - Line 87: there should be a comma after 'employed'
 - Line 91 and 92: the compound labels in the main text and Figure 4 are not consistent
 - Line 106: delete 'almost'
 - Line 110: suggested revised wording, here: "(generally prepared and used in situ throughout this manuscript)"
 - Line 124: 'epimerization' should be 'epimerisation' in order to be consistent with the rest of the text
-
- Page S4: the proton number is not right for compound 1b
 - Page S5, in general procedure A: '0,2 M' should be '0.2 M'
 - Page S5, in general procedure B: '0,3 M' should be '0.3 M'
 - Page S6, compound 3b: '7.60 -7.56' should be '7.60-7.56'
 - Page S15: '0. 017 mmol' should be '0.017 mmol'
 - Page S15: 'hexane:EtOAc = . 4:1 ratio' is incorrect

Reviewer #3 (Remarks to the Author):

Maulide and co-workers describe a new method for the synthesis of alkenes through the chelotropic cycloreversion of N-iminyl aziridines, which were accessed from the reaction of sulfoxonium ylides with hydrazones. The method is quite general with respect to the aldehyde (as the aziridine precursor component) with a variety of aromatic, aliphatic, and vinylic-substituted aldehydes working well (yields of alkene generally in the 50-89% range) after being converted to aziridine starting material. The compatibility of the new methodology with a sensitive α -chiral aldehyde (Figure 6) is particularly noteworthy (compared to the Wittig reaction's tolerance), with virtually no racemization observed. A couple of preliminary results on stereoselective olefinations through utilization of ethyl- and butyl-substituted sulfoxonium ylides are also presented.

Reservations about the report include: I would like to have seen more of an exploration of stereoselectivity in disubstituted alkene formation, with 5-6 examples examining different aspects. For example, how do aliphatic aldehyde-derived and vinylic aldehyde-derived substrates perform in stereoselective olefinations? Is higher E/Z-selectivity obtained with an ortho-substituted aromatic aldehyde or isobutyraldehyde aziridine precursor? Perhaps, the lower reactivity of ethyl or n-butyl-substituted sulfoxonium ylides would prevent more sterically hindered aldehydes from being catered for by the method? Why is there a significant decrease in E/Z selectivity on going from ethyl to n-butyl as a substituent on sulfoxonium salt (Figure 5)? Were the intermediate aziridines isolated and assayed for stereopurity (cis vs trans)? An answer to that question would shed light on whether the high E:Z selectivity for compound 6a is a result of equilibration(isomerization) in a stepwise decomposition mechanism or otherwise. More generally, answers to some of those questions would allow for a more complete comparison with Wittig/HWE methodological scope and also address mechanistic questions.

Background references for successful sulfonium- and sulfoxonium ylide-mediated reactions should be given (page 4, line 59 and line 61): For example, background to sulfoxonium ylide reactions should include citations for Johnson-Corey-Chaykovsky reaction, i.e. cyclopropane formation, epoxide formation, and aziridine formation, etc. Precedent for formation of aziridines through the reaction of sulfoxonium ylides with imines in particular should be mentioned.

Other comments that should be addressed before resubmission or submission elsewhere:

1. Typo: delete " " in 1950's (line 22, page 2)
2. Typo: insert 'p' in middle of 'phoshomium' (line 33, page 2).
3. Typo: fix the spelling of 2-naphthaldehyde (line 53, page 4).
4. Figure 1 (page 3): Show flow of electrons with curly arrows for cheletropic cycloreversion. Also show formation of diazo-compound as byproduct with offshoot arrow from main arrow going to alkene.
5. Background references for some successful sulfonium and sulfoxonium ylide-mediated reactions should be given (page 4, line 59 and line 61).
6. Table 1 on Page 5: the Table heading 'Nucleophile' should be replaced with a term more in keeping with the generic structure MCH₂LG over 2nd reaction arrow, e.g. CH₃LG or pronucleophile.
7. Table 1 on page 5: the generic hydrazone structure over 1st reaction arrow should have 1a-1c listed underneath, as 1a-1c are mentioned repeatedly in subsequent figures throughout the paper.
8. 'Methods' section in paper (line 128 onwards) and in the SI should indicate whether the reaction was heated to 90 degree C or 150 degree C in a sealed flask or otherwise (indicate the type of flask used).
9. Typo: Line 207 on page 13: fix spelling of '1,2-Disubstituted'
10. The SI should mention the source/reference for preparation of triethylsulfoxonium and tributylsulfoxonium salt.
11. In the SI some characterization data (e.g. for compounds 6a, 6b) have fewer ¹³C signals reported than expected, perhaps due to overlapping signals, e.g. For 6a: 13 ¹³C signals expected, only 11 reported; For 6b: 15 ¹³C signals expected, only 13 reported. This needs to be resolved.

On balance, given the underdeveloped nature of the studies on stereoselective olefin formation, which are likely to be the most impactful aspect of this work, I recommend that the authors consider submitting this manuscript to a more specialized organic chemistry journal. However, if the concerns and comments expressed above are addressed, then the manuscript could be resubmitted to this journal.

We are grateful to the reviewers for their insightful comments. We have addressed all the suggestions and comments and hope that the present revised version will be suitable for publication.

Reviewer #1 (Remarks to the Author):

The authors have developed a novel method for olefin formation starting from various aldehydes. The aldehyde reacts with a hydrazone to form an azine, which is then reacted with a sulfoxonium ylide to yield an N-iminyl aziridine. The aziridine undergoes cheletropic elimination of a diazoalkane to give alkene product. Mono and di-substituted alkenes were synthesized in moderate to high yields. The method is tolerant of amides, ethers, esters, alkenes (despite propensity of enals themselves to undergo conjugate addition with sulfoxonium ylides), nitro group, diphenylamino, and aryl halides.

Various aldehydes were employed – aromatic aldehydes with different electron withdrawing & donating substituents, enals, and some representative aliphatic aldehydes, including one with a stereogenic centre α to the carbonyl. Essentially no loss of enantiopurity at the stereogenic centre was observed in the latter reaction. Since C—C and C=C bond forming reactions are of paramount importance in organic chemistry, this submission seems to me to be a valuable addition to the existing bond forming methodologies, and improves on the state of the art by addressing certain shortcomings of other methods that nominally achieve the same C=C bond formations.

The authors claim that their method is the first example of a novel concept – olefination of a carbonyl compound via a 3-membered ring intermediate (as opposed to well-known methods that involve 4- and 5-membered intermediates). However, the numerous variations on the Barton-Kellogg reaction, known for approximately 50 years, are based on exactly this concept, with the 3-membered ring in this case being a thiirane. The thiirane can be formed starting from an azine + hydrogen sulphide, or from a diazoalkane + thiocarbonyl compound. The azine and the diazoalkane, in the different variants, are formed from carbonyl compounds. Extrusion of sulphur from the thiirane is brought about by various means. Treatment with a phosphine also yields the olefin product, presumably through a thiaphosphetane intermediate (i.e. in this instance, it is a 4-center process). There are obvious parallels between the above precedents and the method in the submitted article – certainly, reference should be made to these precedents, with the addition of several literature citations. Most of the salient references can be found in OBC 2005, 3, 28. Other relevant references include Chem. Commun. 1972, 4, and references therein. Since there is clear precedent for the concept of olefination of a carbonyl group through a 3-membered ring intermediate, the authors should acknowledge this.

It would also be appropriate to add references to several papers of Shi-Kai Tian to the introduction, e.g. Chem. Commun. 2011, 47, 2158; JACS 2010, 132, 5018; EJOE 2011, 1084.

We acknowledged the reliable Barton-Kellogg reaction (ref. 20), we further modulated appropriately the text to clarify more in detail the precedent work in this area (page 2 line 20, ref 19-22) and we referenced relevant work by Tian on azaphosphatanes (page 2, line 18, ref. 17-18).

Although there are several similarities to the Barton-Kellogg method, the present method is nonetheless an ingenious departure from the existing state of the art in olefin formation in terms of scope and the exact means through which olefins are formed. Its reasonably general applicability adds significantly to this value. The present method is complementary to the Barton-Kellogg method for olefin formation in the sense that it can be readily used to produce mono or disubstituted olefins, which may not be possible with the Barton-Kellogg method. It may also be complementary to the Wittig reaction (although more examples would be needed to show this unequivocally) in that it appears to yield E-alkenes with alkyl substituents where the Wittig reaction (as it is typically applied) would give Z-alkenes.

In order to increase the general applicability of the method, it would be interesting to see at least one example of formation of a trisubstituted alkene, perhaps by using a sulfoxonium ylide with a branched nucleophilic carbon. I suggest attempting this reaction and including the results of the experiments (whether successful or not) in the present paper. An attempt should also be made to synthesize a disubstituted alkene starting from an aliphatic aldehyde (i.e., Fig 5 with an aliphatic aldehyde).

Attempts at preparing trisubstituted olefins were made (one with a branched nucleophile as suggested, and two starting from ketones) but met with failure (Figure 4.C; 6a,6b,6c). They are now included in the text as unsuccessful substrates. Moreover the substrates scope was considerably enlarged including different examples of disubstituted olefins (Figure 5; 7b, 7d, 7e, 7f).

The authors refer to optimisation of the reaction conditions (Table 1 of article). It is not entirely clear to me why more than 1 equivalent of the trimethylsulfoxonium ylide should be required (i.e. it seems in principle that 1.05 equivalents of sulfoxonium salt + 1.0 equivalent of base should suffice). If the authors have not tried this set of conditions, they should do so and include details of their observations in the article. If reactions under these conditions have already been attempted, details of the results should be included – in particular if there is a decrease in the yield.

An entry with 1.05 equiv. of base and nucleophile was added (figure 2, entry 10), but the substantially diminished yield strongly supports the need for excess amounts of reagents.

For disubstituted olefins (Fig 5), was the ratio of the isomers checked prior to subjecting the product to column chromatography? Isomerization seems unlikely for the alkenes synthesized in this article, but it would be worth checking the ratio of the isomers in the crude product prior to chromatography to be sure that the ratios quoted really are indicative of the selectivity from the reaction rather than an artefact of how the product was purified.

The ratios are now shown in the paper are calculated on crude products mixtures ¹H-NMRs, detailed info are added in the SI.

Was there any evidence of addition of the sulfoxonium ylide to the more substituted end of the azine? Particularly in cases where the yields were low? In cases in which low yields were obtained, what was

the major side-product? Was there a lot of starting material remaining? What caused the low yields in those cases – e.g. any of the entries of Fig 3? Please insert comments on the identities of side products, or if apparently unreacted starting material remained at the end of the reactions that gave relatively low yields.

In no case have we observed addition of the ylide nucleophile at the other carbon terminus of the azine; this is in line with the higher reactivity of the aldimine-like carbon rather than the ketimine-like one. Further support for the lack of reactivity of this type of center comes from the inability of sulfoxonium ylides to olefinate azines derived from ketones. The only side product consistently observed in different substrates is the homologated aldehyde, with no trace of starting material observable (page 6, line 5).

The authors should include a diagram and some discussion in which it is explicitly shown that the initial cheletropic elimination product is diazoalkane, and in which an explanation is given for what happens to this product subsequently. This information, implied in the submitted article, may not be obvious to readers who are not familiar with the area.

We did not pursue a clear identification of the diazoalkane side product decomposition pathway. Strong evidence of its generation is given by different experiments, but we believe that further investigations in this context would fall outside the scope of this manuscript.

Comments on Mechanistic Study

Unless I am mistaken, no details of the experiment in which 8 is reacted with DMSO are included in the experimental section. A description of this experiment with NMR spectra should be included. What products, aside from the 43% that formed benzophenone, were formed in this process?

By mistake the experimental data for this experiment was not included, now it is included with a NMR description. Since we were interested in proving the possible intermediacy of a diazoalkane, we pursued the product of its oxidation. Other side products eventually formed would not necessarily be relevant to the methodology presented herein, because their formation under this set of conditions does not necessarily imply their formation in the standard set of conditions. On the contrary, the formation of the diazo compound via cheletropic reversion is, so this was by far the most important compound we sought to observe.

The mechanism of the decomposition of 8 needs further clarification – the evidence already presented, although consistent with the authors' rationale, is not sufficiently comprehensive. The yield of the reaction of 8 with DMSO is somewhat low and, moreover, no details are given about this experiment in the supporting information. Use of commercially available 4,4'-bis(N,N-dimethylamino)benzophenone in a similar reaction sequence to that given in the submission would lead to an iminyl aziridine and hence a diazo diarylmethane with two very electron rich aryl groups, which may give a high yield in an oxidation reaction with DMSO based on reference 31 from the submitted article. Alternatively, 4,4'-bis(methoxy)benzophenone could be used. A high yield in a reaction such as this would show that a diazoalkane is indeed formed by cheletropic elimination from the iminyl aziridine.

Is there evidence for the formation of dimethylsulfide as by-product in the reaction of 8 with DMSO? If not, what else is formed as a by-product?

The experimental part was added. Moreover, we believe that the yield of 43% registered for the benzophenone formation, while in an absolute sense being somewhat low, is rather significant if one takes into consideration that this is nothing but a side product obtained in conditions originally optimized for the obtention of a completely different compound. Logically, the use of a more electron rich diarylazomethane in the final reaction conditions might lead to increased yields of oxidized product, but this would happen only if the overall sequence (namely azine formation, nucleophilic attack and cheletropic reversion) is not affected by this structural modification, so the optimization led us to the use of benzophenone (figure 2 and figure 4.B). The formation of its derivate, diazomethane, is inferred from a side experiment leading to the oxidized product and further confirmed by experiments aimed at trapping the diazoalkane. These are reported in the text (page 12; eq. 2, eq.3). Lastly, gaseous dimethyl sulfide was not searched for, as we believe that the trapping or detection of this gas at 120°C is out of the scope of the methodology.

Comments on Reactions with Aliphatic Aldehydes

It is stated on page 7 of the article that "It is likely that 1b, carrying an extended π -system, stabilizes the hydrazone tautomer". Use of hydrazones derived from substituted benzophenones to form azines derived from aliphatic aldehydes might allow the above supposition to be tested.

Other substituted benzophenone could favor, in principle, the hydrazone tautomer. Others were actually tested and the results are now reported (figure 4.B).

An azine derived from commercially available 4-nitrobenzophenone and an aliphatic aldehyde should favour the hydrazone tautomer, and thus lead to higher yields in reactions with aliphatic aldehydes. The authors should consider investigating this possibility.

We added this investigation into our methodology (figure 4.B, 1e), but the reaction was met with overall failure. As added in the text, the azine formation slows down considerably and the cheletropic reversion itself provides only trace olefin, with considerable degradation of the material happening.

An azine derived from 4,4'-bis(N,N-dimethylamino)benzophenone and an aliphatic aldehyde should favour the enamine tautomer due to the strongly electron donating character of the dimethylamino groups.

In the event, use of an electron-rich azine essentially prevents any successful aziridine formation (figure 4.B, 1d).

Comments on Supporting Information

The numbering of the iminyl aziridine, and the cyclopropane product (8 and 9, respectively, in the main article) is not consistent between the main article and supporting information – see pg 14 of Supporting Information.

Synthesis of hydrazone 1b (pg 4) – why does the description say "ketone or aldehyde"? Surely only benzophenone was used?

1b is incorrectly labelled as a hydrazine rather than a hydrazone in the supporting information (pg 4)

These typing errors were corrected.

The reactions were all carried out under anhydrous conditions. How exactly was the filtration of the azine solution carried out? If it was done under inert atmosphere, please provide more detail on the method of filtration used.

More detailed data were included in the SI.

To what extent do the authors think water affects the yields of the reactions? It is clearly a factor, since the reactions were carried out under anhydrous conditions. Could the lower yields in some cases be a result of the intervention of water? Perhaps activated molecular sieves could be used in place of MgSO₄ to increase yields – 3 Å sieves are suitable for drying acetonitrile (see JOC 2010, 75, 8351).

Experiment proves that water is detrimental to the reaction outcome (figure 2, entry 11), though not completely shutting it down. So adventitious water could lower the yield but we exclude any major lowering of the yield (i.e. variations in the 5% range). Finally, MgSO₄ was used in CH₂Cl₂ or MeOH for the condensation step and it was chosen after some optimization and screening as the most performing/best yielding additive. On the other hand we use acetonitrile for the subsequent steps, which is stored over 3 Å mol. sieves as standard. The use of a drying agent in these steps was found not to be particularly beneficial, so we moved on in the optimized conditions.

Does the formation of the azine have to be done in CH₂Cl₂? Perhaps it could be done in CH₃CN, followed by filtration to remove MgSO₄, removing the need for solvent removal after formation of the azine. If this is not possible, the authors should comment on why the use of CH₂Cl₂ for the azine formation is essential, either in the Supporting Information or in the main article.

Different conditions for the condensation were tested, though not reported as we believe they fall outside the scope of the paper. Turns out that CH₂Cl₂ or MeOH in the optimized conditions, respectively are the most suitable solvent as far as it goes for yields, reaction times and easiness of operations. Solvent removal or addition is anyway needed due to different concentration in the subsequent reaction steps.

Comments on the Stereoselectivity of Disubstituted Alkene Formations

Significant elaboration on the reasons for the observed E-stereoselectivity are required. At the outset, and as stated elsewhere, the isomer ratios of the crude products should be checked.

The authors state that the E-selectivity may be a result of "stereospecific cheletropic elimination of the more stable trans-aziridine". At the very least, this sentence needs to be re-formulated, as it implies that the aziridine itself is eliminated (i.e., the phrasing should be made more precise). However, a more serious issue is what the sentence implies about the chemistry. Do the authors think that the formation of the iminyl aziridine is reversible, and that E-selectivity results from elimination from the trans-aziridine being faster than elimination from the cis-aziridine? Or is the formation of the aziridine irreversible, meaning that the predominant E-selectivity is due to faster formation of the E-aziridine? It may be difficult to answer these questions (although the experiments suggested below would help), but a more precise explanation of what the authors believe to be possible is warranted.

More detailed experiments are added and a proposed mechanism based on these is provided (eq. 2-6 and figure 6).

Given that it has been shown to be possible to isolate 8, perhaps the authors should attempt to isolate an iminyl aziridine that goes on to give a vicinally disubstituted alkene (e.g. derived from a triethylsulfoxonium salt and t-BuOK), check its stereochemistry, and then check if the stereochemistry is retained (or otherwise) during the course of decomposition to alkene. This would give a wealth of information about the concertedness, or otherwise, of the second step of the reaction, and about the source of the stereoselectivity. Trapping experiments could also be employed to check if the elimination is not concerted.

More detailed experiments are added and a proposed mechanism based on these is provided (eq. 2-6 and figure 6).

There are several examples of typographical errors, for example

Phosphorus – main article pg 13

triethylsulfoxonium iodide - ESI pg 13

Please fix all such errors. Use "room temperature" in place of rt in main article and supporting information. Also "crude" is not a noun – use "crude product" (experimental sections).

These typos had been addressed.

Recommendation

I recommend that this article be accepted once the revisions and additional experiments suggested above have been done. Even if the suggested experiments do not yield the desired results, they may lead to a significantly better understanding of the reaction, its mechanism, and the source of the E-selectivity (where appropriate).

Reviewer #2 (Remarks to the Author):

This manuscript by Maulide and co-workers describes a method for olefination of aldehydes via sequential N-iminyl aziridine formation followed by cheletropic cycloreversion. The proposed mechanism involves formation of a three-membered aziridine intermediate using sulfoxonium iodide as the methylenide source with subsequent thermal-induced elimination of a diazo species to reveal the olefin.

Conceptually, the reaction is interesting, as it represents a new mechanistic strategy for carbonyl olefination that is distinct from the traditional approaches in Figure 1. Practically speaking, the reaction uses straightforward reaction conditions, the yields are typically good, and the functional group tolerance is reasonably well established. There are some notable limitations of the method. First, the substrate scope appears to be somewhat narrow; yields with aliphatic aldehydes are in the 50–70% range, and no examples are provided with ketones to prepare 1,1-disubstituted olefins. Only two examples are provided of non-methylenide alkylidene transfer to prepare 1,2-disubstituted alkenes. In the context of multi-step organic synthesis, it is unclear what advantages this method might offer compared to existing methods.

The Supporting Information is well organized, detailed, and complete. All new compounds are thoroughly characterized. Aside from the points below, the conclusions are generally supported by the data, and the manuscript is presented in a scholarly manner. Based on the overall conceptual novelty, I recommend that this manuscript be published in Nature Communications, following incorporation of the following minor revisions:

(1) For examples of non-methylenide alkylidene transfer to prepare 1,2-disubstituted alkenes, the levels of E selectivity appear to be variable based on the two examples that are provided (>20:1 in one case, 4:1 in the other). More 1,2-disubstituted examples are necessary to make general conclusions about the stereoselectivity of this transformation.

New substrates have been added, to address this shortcomings; new non-methylenide type olefination are now included (figure 4.C and figure 5).

(2) In Ref. 30, which is cited in the text accompanying Fig. 6, the two Wittig-type examples that the authors refer to with this specific substrate involve epimerization at the level of 95% to 84% ee and 95% to 80% ee. The authors describe this as "significant erosion". Given that the descriptor 'significant' is subjective in this context—I imagine some chemists would also call this "modest epimerization"—I would prefer that the actual numbers from this earlier report be presented here to make comparison more straightforward for the reader.

The equation 1 had been made clearer to show the comparison between different methods.

(3) In the figures, it is unclear at first-glance whether the yields correspond to final yield over all three steps or only over the final two steps. From the way that the experimental procedure is written, the former appears to be the case, but it would be helpful to the reader if this were clarified in the footnotes.

Clearer definition of the yield calculation had been stated both in the main text (figures' legenda and in the SI.

(4) A few minor typographical, stylistic, and grammatical issues were present throughout the manuscript and Supporting Information

- Line 8: "an unsaturation" is ambiguous in this context. There may be a missing word here.
 - Line 47: insert a comma after 'project'
 - Line 53: '2-napthaldheyde' should be '2-napthaldehyde'
 - Line 69: there should be a hyphen between 'electron' and 'neutral'
 - Line 71: there should be a comma after 'conditions'
 - Line 78 and 79: 'unsaturred' should be 'unsaturated'
 - Line 81: suggested revised wording, here: "...analysis, despite the fact that reaction between sulfoxonium ylides and α,β -unsaturated carbonyls is a textbook transformation."
 - Line 87: there should be a comma after 'employed'
 - Line 91 and 92: the compound labels in the main text and Figure 4 are not consistent
 - Line 106: delete 'almost'
 - Line 110: suggested revised wording, here: "(generally prepared and used in situ throughout this manuscript)"
 - Line 124: 'epimerization' should be 'epimerisation' in order to be consistent with the rest of the text
-
- Page S4: the proton number is not right for compound 1b
 - Page S5, in general procedure A: '0,2 M' should be '0.2 M'
 - Page S5, in general procedure B: '0,3 M' should be '0.3 M'
 - Page S6, compound 3b: '7.60 –7.56' should be '7.60–7.56'
 - Page S15: '0. 017 mmol' should be '0.017 mmol'
 - Page S15: 'hexane:EtOAc = . 4:1 ratio' is incorrect

These typos have been addressed.

Reviewer #3 (Remarks to the Author):

Maulide and co-workers describe a new method for the synthesis of alkenes through the cheletropic cycloreversion of N-iminyl aziridines, which were accessed from the reaction of sulfoxonium ylides with hydrazones. The method is quite general with respect to the aldehyde (as the aziridine precursor component) with a variety of aromatic, aliphatic, and vinylic-substituted aldehydes working well (yields

of alkene generally in the 50-89% range) after being converted to aziridine starting material. The compatibility of the new methodology with a sensitive α -chiral aldehyde (Figure 6) is particularly noteworthy (compared to the Wittig reaction's tolerance), with virtually no racemization observed. A couple of preliminary results on stereoselective olefinations through utilization of ethyl- and butyl-substituted sulfoxonium ylides are also presented.

Reservations about the report include: I would like to have seen more of an exploration of stereoselectivity in disubstituted alkene formation, with 5-6 examples examining different aspects. For example, how do aliphatic aldehyde-derived and vinylic aldehyde-derived substrates perform in stereoselective olefinations? Is higher E/Z-selectivity obtained with an ortho-substituted aromatic aldehyde or isobutyraldehyde aziridine precursor? Perhaps, the lower reactivity of ethyl or n-butyl-substituted sulfoxonium ylides would prevent more sterically hindered aldehydes from being catered for by the method? Why is there a significant decrease in E/Z selectivity on going from ethyl to n-butyl as a substituent on sulfoxonium salt (Figure 5)? Were the intermediate aziridines isolated and assayed for stereopurity (cis vs trans)? An answer to that question would shed light on whether the high E:Z selectivity for compound 6a is a result of equilibration(isomerization) in a stepwise decomposition mechanism or otherwise. More generally, answers to some of those questions would allow for a more complete comparison with Wittig/HWE methodological scope and also address mechanistic questions.

More substrate to address the issue E/Z ratio were tested (figure 5), moreover the pure *cis* or *trans* aziridines have been insolated an used separately, showing the sterospecificity of the reaction (eq.4 and eq. 5).

Background references for successful sulfonium- and sulfoxonium ylide-mediated reactions should be given (page 4, line 59 and line 61): For example, background to sulfoxonium ylide reactions should include citations for Johnson-Corey-Chaykovsky reaction, i.e. cyclopropane formation, epoxide formation, and aziridine formation, etc. Precedent for formation of aziridines through the reaction of sulfoxonium ylides with imines in particular should be mentioned.

Background references on chemistry of sulfoxonium ylides and aziridines has been added (ref.XX,XX, XX)

Other comments that should be addressed before resubmission or submission elsewhere:

1. Typo: delete " ' " in 1950's (line 22, page 2)
2. Typo: insert 'p' in middle of 'phoshomium' (line 33, page 2).
3. Typo: fix the spelling of 2-naphthaldehyde (line 53, page 4).
4. Figure 1 (page 3): Show flow of electrons with curly arrows for cheletropic cycloreversion. Also show formation of diazo-compound as byproduct with offshoot arrow from main arrow going to alkene.
5. Background references for some successful sulfonium and sulfoxonium ylide-mediated reactions

should be given (page 4, line 59 and line 61).

6. Table 1 on Page 5: the Table heading 'Nucleophile' should be replaced with a term more in keeping with the generic structure MCH₂LG over 2nd reaction arrow, e.g. CH₃LG or pronucleophile.
7. Table 1 on page 5: the generic hydrazone structure over 1st reaction arrow should have 1a-1c listed underneath, as 1a-1c are mentioned repeatedly in subsequent figures throughout the paper.
8. 'Methods' section in paper (line 128 onwards) and in the SI should indicate whether the reaction was heated to 90 degree C or 150 degree C in a sealed flask or otherwise (indicate the type of flask used).
9. Typo: Line 207 on page 13: fix spelling of '1,2-Disubstituted'
10. The SI should mention the source/reference for preparation of triethylsulfoxonium and tributylsulfoxonium salt.
11. In the SI some characterization data (e.g. for compounds 6a, 6b) have fewer ¹³C signals reported than expected, perhaps due to overlapping signals, e.g. For 6a: 13 ¹³C signals expected, only 11 reported; For 6b: 15 ¹³C signals expected, only 13 reported. This needs to be resolved.

These issues had been addressed

On balance, given the underdeveloped nature of the studies on stereoselective olefin formation, which are likely to be the most impactful aspect of this work, I recommend that the authors consider submitting this manuscript to a more specialized organic chemistry journal. However, if the concerns and comments expressed above are addressed, then the manuscript could be resubmitted to this journal.

REVIEWERS' COMMENTS:

Reviewer #1 (Remarks to the Author):

On the basis that the authors have incorporated most of the changes that the other reviewers and I suggested, I recommend publication of this article. However, there still remain several minor issues that I think should be addressed prior to publication. I detail these below.

I would like to see details of the attempted preparations of 6a, 6b and 6c included in the ESI - it is important to know what procedures have been tried and not worked.

In their rebuttal letter, the authors refer to synthesis of a compound F, which I do not see in the paper. Was this just a typo in the letter, or has there been a compound omitted?

Contrary to the opinion of the authors, I believe that trapping of dimethylsulfide in the experiment involving compound 9 would be straightforward - all that would be required would be a liquid nitrogen trap attached to the reaction flask. However, this is not an essential experiment in the context of this submission.

The experiments with the stereo-defined cis & trans aziridines are very illustrative. This is a nice addition by the authors.

The yield for eq 6 is rather low, and I don't really understand the discussion relating to it at the bottom of pg 11. I think the authors need to make more clear what they mean in this section.

Numbers on pg 11 of the article document need to be updated in the revised manuscript (i.e. 8 should be 9 etc.).

Finally, there are quite a few grammatical and typographical errors that have been introduced in the updated submission. I would address these prior to publication.

Reviewer #2 (Remarks to the Author):

In the revised version of the manuscript by Maulide and coworkers, the authors have addressed the issues that I raised regarding the earlier submission; as a result, scientific and technical aspects of the manuscript are now improved. I am in support of publication in Nat. Commun. after the following minor points have been addressed.

(1) While the authors fixed most of the typographical, stylistic, and grammatical issues that I raised initially, there were several additional issues that were introduced during revision (or that I missed the first time around):

- Line 8: "an unsaturation" is ambiguous in this context. Perhaps they mean "a unit of unsaturation." (This was pointed out last time around not fixed during revision.)
- Line 39: "ring, as does recently developed..."
- Line 46: delete comma after 'aziridine'
- Line 52: "two aforementioned challenges"
- Line 66: "rigorous exclusion of water while beneficial is not mandatory"
- Line 82: "Isolated yields over two steps"
- Line 99: "hydrazones with more electron-rich (1d) and electron-poor (1e) aryl substituents were tested but failed in providing the desired product. Compound 1e exhibited prohibitively slow rate of azine formation and provided trace ...degradation. Compound 1d led to efficient azine formation, but nucleophilic attack from the sulfoxonium ylide was not observed."
- Line 110: 'Figure 5' should be 'Figure 4'

- Line 123 (equation 1): It would be helpful if references were present in this figure. Also, ">80%" is unclear. In the original paper, this ee value is reported as 80%, so it's unclear what the authors mean here.
- Line 124: throughout this page, the compounds are mis-numbered. Compound 8 should be 9, and 9 should be 10.
- Line 128: "These results provide"
- Line 141, eq. 6: the reagent used in the second step should be Et₃S(I) instead of Me₃S(I).
- Line 142: 'Figure 6' should be 'Figure 5'
- Line 144; 'collapses
- Line 146: "faces of attack" and later "3-membered ring"
- Line 147: "3-membered ring"
- Line 148: "geometric interconversion"
- Line 149: "preference for attack on one of the two faces"
- Line 150: 'cycloreverts'
- Line 153: "compound. This accounts for..."
- Line 172: insert 'was' before "stirred at"
- Line 186: This should be two sentences, as in the previous instance. "...EtOAc (x3). The combined..."
- Supporting Information: The J values are not described in a consistent format. For example, Page S7, compound 3e, 'J' should be italicized. Page S8, compound 3f, two spaces are missing next to the equals sign. (This can be corrected with a Ctrl + F "J =" and "J=".)
- Supporting Information: The font sizes are not unified in the SI. For example, Page S18, the chemical shifts of ¹³C NMR are in a small size. It is recommended that the authors recheck the entire Supporting Information document for these types of formatting issues.

Reviewer #3 (Remarks to the Author):

The authors have addressed my previous comments in a satisfactory manner, especially with regard to the examination of internal alkene scope. I recommend publication without any further revisions.

REVIEWERS' COMMENTS:

Reviewer #1 (Remarks to the Author):

On the basis that the authors have incorporated most of the changes that the other reviewers and I suggested, I recommend publication of this article. However, there still remain several minor issues that I think should be addressed prior to publication. I detail these below.

I would like to see details of the attempted preparations of 6a, 6b and 6c included in the ESI - it is important to know what procedures have been tried and not worked.

The procedure used experimentally in the attempted preparations of 6a, 6b and 6c have been included in the SI (page 1 line 32 and pages 12-13-21-21), comprising of complete characterization were necessary.

In their rebuttal letter, the authors refer to synthesis of a compound F, which I do not see in the paper. Was this just a typo in the letter, or has there been a compound omitted?

This was a typo since no letter has been attributed to any compound throughout the manuscript or its preparation.

Contrary to the opinion of the authors, I believe that trapping of dimethylsulfide in the experiment involving compound 9 would be straightforward - all that would be required would be a liquid nitrogen trap attached to the reaction flask. However, this is not an essential experiment in the context of this submission.

The experiments with the stereo-defined cis & trans aziridines are very illustrative. This is a nice addition by the authors.

The yield for eq 6 is rather low, and I don't really understand the discussion relating to it at the bottom of pg 11. I think the authors need to make more clear what they mean in this section.

We agree that the yield is rather low, from the optimization table 1 in the main text we show that simple sulfonium salts are not useful reagents for this chemistry. Nevertheless, the aziridine formed by action of the two different reagent should be the same, this is an important data for understanding the mechanism and the key steps in the reaction. This is better explained now in the text (page 11 lines 12-16)

Numbers on pg 11 of the article document need to be updated in the revised manuscript (i.e. 8 should be 9 etc.).

The numbering should now be consistent.

Finally, there are quite a few grammatical and typographical errors that have been introduced in the updated submission. I would address these prior to publication.

We addressed the typographical errors.

Reviewer #2 (Remarks to the Author):

In the revised version of the manuscript by Maulide and coworkers, the authors have addressed the issues that I raised regarding the earlier submission; as a result, scientific and technical aspects of the manuscript are now improved. I am in support of publication in Nat. Commun. after the following minor points have been addressed.

(1) While the authors fixed most of the typographical, stylistic, and grammatical issues that I raised initially, there were several additional issues that were introduced during revision (or that I missed the first time around):

- Line 8: "an unsaturation" is ambiguous in this context. Perhaps they mean "a unit of unsaturation." (This was pointed out last time around not fixed during revision.)
- Line 39: "ring, as does recently developed..."
- Line 46: delete comma after 'aziridine'
- Line 52: "two aforementioned challenges"
- Line 66: "rigorous exclusion of water while beneficial is not mandatory"
- Line 82: "Isolated yields over two steps"
- Line 99: "hydrazones with more electron-rich (1d) and electron-poor (1e) aryl substituents were tested but failed in providing the desired product. Compound 1e exhibited prohibitively slow rate of azine formation and provided trace ...degradation. Compound 1d led to efficient azine formation, but nucleophilic attack from the sulfoxonium ylide was not observed."
- Line 110: 'Figure 5' should be 'Figure 4'
- Line 123 (equation 1): It would be helpful if references were present in this figure. Also, ">80%" is unclear. In the original paper, this ee value is reported as 80%, so it's unclear what the authors mean here.
- Line 124: throughout this page, the compounds are mis-numbered. Compound 8 should be 9, and 9 should be 10.
- Line 128: "These results provide"
- Line 141, eq. 6: the reagent used in the second step should be Et₃S(I) instead of Me₃S(I).
- Line 142: 'Figure 6' should be 'Figure 5'
- Line 144; 'collapses'
- Line 146: "faces of attack" and later "3-membered ring"
- Line 147: "3-membered ring"
- Line 148: "geometric interconversion"
- Line 149: "preference for attack on one of the two faces"

- Line 150: 'cycloreverts'
- Line 153: "compound. This accounts for..."
- Line 172: insert 'was' before "stirred at"
- Line 186: This should be two sentences, as in the previous instance. "...EtOAc (x3). The combined..."
- Supporting Information: The J values are not described in a consistent format. For example, Page S7, compound 3e, 'J' should be italicized. Page S8, compound 3f, two spaces are missing next to the equals sign. (This can be corrected with a Ctrl + F "J =" and "J=".)
- Supporting Information: The font sizes are not unified in the SI. For example, Page S18, the chemical shifts of ¹³C NMR are in a small size. It is recommended that the authors recheck the entire Supporting Information document for these types of formatting issues.

We addressed the typographical errors.

Reviewer #3 (Remarks to the Author):

The authors have addressed my previous comments in a satisfactory manner, especially with regard to the examination of internal alkene scope. I recommend publication without any further revisions.